

# Development of EST-SSR markers based on transcriptome for genetic analysis in *Radix Ardisia*

Deqiang Ren, Wenwen Wu, Qinqin Wen and Yujiao Li

College of Pharmacy, Guizhou University of Traditional Chinese Medicine, Guiyang, Guizhou, China

## ABSTRACT

*Radix Ardisia* is a commonly used medicine for the Miao nationality distributed over Guizhou and used to treat laryngeal diseases. The medicinal materials of the *Radix Ardisia* come from various sources, including Bailiangjin (*Ardisia crispa* (Thunb.) A. DC.), Zhushagen (*Ardisia crenata* Sims), Hongliangsan (*Ardisia crenata* Sims var. bicolor (Walker) C. Y. Wu et C. Chen), Xibingbailiangjin (*Ardisia crispa* (Thunb.) A. DC. var. dielsii (Levl.) Walker) and Dayebailiangjin (*Ardisia crispa* (Thunb.) A. DC. var. amplifolia Walker). With the continuous improvement of the medicinal value of the Miao medicine and the increasing scarcity of wild resources, it is of great practical significance to solve the problems of effective identification and genetic structure analysis of medicinal materials and adulterants, for the protection of its germplasm resources and the protection of clinical drug safety. The development of expressed sequence tag-simple sequence repeat (EST-SSR) based on transcriptome strategy is considered to be a very effective means. In this study, 51,237 sequences of *A. crenata* were retrieved, with a total length of 71.4 MB. A total of 32,827 SSR loci were detected, averaging one SSR locus per 2.1 KB. The distribution of primers was detected, and 28,322 SSR loci were unigene SSR. 32,827 pairs of EST-SSR molecular markers were developed for the whole genome, with an average of 0.64 pairs of SSR primers for each unigene, and the sequence coverage was high. The statistical analysis showed that six types of SSR nucleotides could be detected, but the number and frequency of EST-SSR in different primitive types were significantly different. Mononucleotide and dinucleotide were the main repeat types, accounting for 90.51% of the total SSR of *A. crenata*. Sixty pairs of primers were randomly selected and applied to genetic research. Among them, 51 pairs could amplify 200 polymorphic bands. Genetic analysis was carried out on 46 mixed species of *Radix Ardisia*. The results showed that the original plants of *Radix Ardisia* showed high genetic diversity and could be divided into two populations. The results of systematic clustering showed that the EST-SSR used could well distinguish *Radix Ardisia* and its easily mixed species; it can be applied to the identification of *Radix Ardisia*, as well as the molecular identification between the original Bailiangjin, Zhushagen and Hongliangsan. This study can provide a reference for the genetic analysis of the *Radix Ardisia*.

Corresponding author
Deqiang Ren, 343519828@qq.com

## INTRODUCTION

*Radix Ardisia* is a commonly used medicine of the Miao people in Guizhou Province. It is regarded as a good throat medicine by the Miao people and is the main component of the Miao medicine's proven prescription (*Zhao & Du, 2016*; *Wu, 2012*). The primary source of the medicinal materials of *Radix Ardisia* is wild. Due to the lack of effective identification methods, the types and contents of the effective ingredients in therapeutic components from different origins are quite different (*Wang et al., 2020*), and the quality of the medicinal materials is difficult to guarantee. With the continuous improvement of therapeutic value and the increasing scarcity of wild resources, it is urgent to establish an effective identification method, carry out an in-depth analysis of its genetic structure, clarify the origin and genetic relationship, and carry out research on the breeding practice, which is of great significance for the protection of germplasm resources and the safety of clinical medicine.

Due to the lag of genetic research on *Radix Ardisia*, there are few reports on the genetic diversity analysis. *Jiang et al. (2006)*, *Yuan et al. (2024)*, and *Tao, Liao & Luo (2011)* use molecular markers to study the genetic diversity of *Radix Ardisia* and found that the genetic polymorphism within *Radix Ardisia* is high and the variation is rich. *Zhang, Guo & Yang (2011)*, *Wang (2012)* and *Chen et al. (2017)* used sporopollen characteristics and DNA barcode technology to identify the species of *Ardisia*, and the results showed that the *matK* sequence can better identify the species of *Ardisia*. However, more data are needed for accurate identification and genetic research of plants within the genus. Expressed sequence tag simple sequence repeat (EST-SSR) markers have the characteristics of simple technology, good repeatability, high polymorphism and codominant inheritance in genome SSR markers but also have the advantages of cheap primer development, good versatility, clear bands and accessible statistics. In addition, as EST-SSR is a part of the coding gene, it can directly obtain the relevant information of gene expression, which may directly identify the alleles that determine critical phenotypic traits. With the development of transcriptomics technology, using its sequence to develop whole genome EST-SSR molecular markers has also become a very simple and effective method, which has been widely used in the identification of medicinal plant germplasm resources, genetic mapping analysis, polymorphism analysis, phylogenetic analysis, etc. in many plants (*Zhang et al., 2023*; *Shi et al., 2023*). *Singh, Sane & Thuppil (2024)* have shown that EST-SSR is an effective means for classification and identification, genetic diversity, core collection screening, *etc.* However, no relevant research has been carried out in the study of *Radix Ardisia*.

Therefore, this study intends to develop genome-wide EST-SSR primers using the transcriptome data of *Ardisia crenata* Sims, apply them to genetic research, analyse the genetic diversity of *Ardisia* species and the genetic structure of the population, to provide a reference for the identification of germplasm resources.

## MATERIALS & METHODS

### Experimental materials

The test materials were collected in Guizhou, Yunnan, Sichuan, Guangxi and other provinces (Table 1). The certificate specimens were identified by Professor Wei Shenghua, the director of the Department of Traditional Chinese Medicine Cultivation of Guizhou University of Traditional Chinese Medicine, and belong to the genus *Ardisia*. The samples and certificate specimens were stored in the Institute of Traditional Chinese Medicine Cultivation (Breeding) and Processing of Guizhou University of Traditional Chinese Medicine.

### Transcriptome sequencing and EST-SSR development

Total RNA was extracted using Trizol reagent and purified using the Oligotex mRNA kit (*Jiang et al., 2011*). Aglient 2100 was used to detect the concentration and integrity of RNA samples, after library construction, transcriptome sequencing was performed using Illumina HiSeq 4000 platform (*Hansen, Brenner & Dudoit, 2011*). After obtaining raw sequencing data from transcriptome sequencing, data filtering was performed to remove low-quality reads that contain high levels of ligands and unknown base N, resulting in high-quality sequencing data (clean reads). Trinity (*Grabherr et al., 2011*) was used to reassemble clean reads, remove identical sequences with Cd-hit, and then use Tgicl for clustering. Sequences were merged with similarity greater than 90% and overlap length greater than 35 to obtain Unigenes. Unigenes were annotated using BLAST and HMMER. GO and KEGG analysis was performed and expression levels were estimated using RSEM (RNA Seq by Expectation Maximization) (*Altschul et al., 1997*; *Davidson & Oshlack, 2014*). FPKM (fragments per kilobase of script per million mapped reads) value was used to represent the expression abundance of corresponding Unigenes.

MISA (*Beier et al., 2017*) (microsatellite identification tool) was used to detect SSR loci of Unigene. SSR data was retrieved and analysed based on the criteria of at least 12, six, and five repetitions for single, double, and triple bases, and no less than four repetitions for four, five, and six bases, respectively. Then, the obtained SSR data was classified and statistically analysed.

### Genetic diversity analysis

Data analysis refer to *Liu et al. (2022)*. For each microsatellite marker, the number of alleles ($Na$), number of effective alleles ($Ne$), observed heterozygosity ($Ho$), expected heterozygosity ($He$), unbiased expected heterozygosity ($uHe$), inbreeding coefficient ($F$), and Shannon's information index ($I$) were calculated using GenAlEx v6.5 (*Peakall & Smouse, 2012*). The polymorphic information content (PIC) was calculated using Power Marker V3.25 software (*Liu & Muse, 2005*). F-statistics calculations (FIS, FIT, and FST) and principal coordinate analysis (PCoA) were also performed in GenAlEx v6.5 combined with Microsoft Excel. A neighbour-joining (NJ) tree was generated based on pairwise genetic distances between individuals using Power Marker V3.25 and plotted with iTOL (*Letunic & Bork, 2021*). The population structure analysis was performed with Bayesian model-based admixture analyses in Structure v2.3.4. We set the number of Markov chain

**Table 1  Martirial of *Radix Ardisia*.**

| Serial number | Identifier | Chinese name | Latin name | Source |
|---|---|---|---|---|
| 1 | BH01 | Baihuazijinniu | *Ardisia merrillii* E. Walker | Huoba Town, Longchuan County, Yunnan Province, China |
| 2 | BH02 | Baihuazijinniu | *Ardisia merrillii* E. Walker | Xuangang Township, Mangshi City, Yunnan Province, China |
| 3 | BH03 | Baihuazijinniu | *Ardisia merrillii* E. Walker | Taiping County, Kunming City, Yunnan Province, China |
| 4 | BH04 | Baihuazijinniu | *Ardisia merrillii* E. Walker | Nongdao Town, Ruili City, Yunnan Province, China |
| 5 | BH05 | Baihuazijinniu | *Ardisia merrillii* E. Walker | Ruili City, Yunnan Province, China |
| 6 | BL01 | Bailiangjin | *Ardisia crispa* (Thunb.) A. DC. | Wandong Bridge, Guiyang City, Guizhou Province, China |
| 7 | BL02 | Bailiangjin | *Ardisia crispa* (Thunb.) A. DC. | Chishui County, Zunyi City, Guizhou Province, China |
| 8 | BL03 | Bailiangjin | *Ardisia crispa* (Thunb.) A. DC. | Sandu County, Qiannan Prefecture, Guizhou Province, China |
| 9 | BL04 | Bailiangjin | *Ardisia crispa* (Thunb.) A. DC. | Huishui County, Qiannan Prefecture, Guizhou Province, China |
| 10 | BL05 | Bailiangjin | *Ardisia crispa* (Thunb.) A. DC. | Anlong County, Qiannan Prefecture, Guizhou Province, China |
| 11 | BL06 | Bailiangjin | *Ardisia crispa* (Thunb.) A. DC. | Qixingguan District, Bijie City, Guizhou Province, China |
| 12 | DF01 | Dongfangzijinniu | *Ardisia elliptica* Thunb. | Guangxi Province, China |
| 13 | GG01 | Jiuguanxue | *Ardisia brevicaulis* Diels | Duyun City, Qiannan Prefecture, Guizhou Province, China |
| 14 | GG02 | Jiuguanxue | *Ardisia brevicaulis* Diels | Dushan County, Qiannan Prefecture, Guizhou Province, China |
| 15 | GG03 | Jiuguanxue | *Ardisia brevicaulis* Diels | Guiyang City, Guizhou Province, China |
| 16 | HL01 | Hongliangsan | *Ardisia crenata* Sims var. bicolor (Walker) C. Y. Wu et C. Chen | Guiyang City, Guizhou Province, China |
| 17 | HL02 | Hongliangsan | *Ardisia crenata* Sims var. bicolor (Walker) C. Y. Wu et C. Chen | Xixiu District, Anshun City, Guizhou Province, China |
| 18 | HL03 | Hongliangsan | *Ardisia crenata* Sims var. bicolor (Walker) C. Y. Wu et C. Chen | Jianhe County, Qiandongnan Prefecture, Guizhou Province, China |
| 19 | HL04 | Hongliangsan | *Ardisia crenata* Sims var. bicolor (Walker) C. Y. Wu et C. Chen | Taijiang County, Qiandongnan Prefecture, Guizhou Province, China |
| 20 | HL05 | Hongliangsan | *Ardisia crenata* Sims var. bicolor (Walker) C. Y. Wu et C. Chen | Yingjiang County, Dehong Prefecture, Yunnan Province, China |
| 21 | HL06 | Hongliangsan | *Ardisia crenata* Sims var. bicolor (Walker) C. Y. Wu et C. Chen | Yingjiang County, Dehong Prefecture, Yunnan Province, China |
| 22 | HL07 | Hongliangsan | *Ardisia crenata* Sims var. bicolor (Walker) C. Y. Wu et C. Chen | Ceheng County, Qiannan Prefecture, Guizhou Province, China |

Ren et al. (2025), *PeerJ*, DOI 10.7717/peerj.19560

**Table 1** (*continued*)

| Serial number | Identifier | Chinese name | Latin name | Source |
|---|---|---|---|---|
| 23 | HL08 | Hongliangsan | *Ardisia crenata* Sims var. bicolor (Walker) C. Y. Wu et C. Chen | Leishan County, Qiandongnan Prefecture, Guizhou Province, China |
| 24 | HL09 | Hongliangsan | *Ardisia crenata* Sims var. bicolor (Walker) C. Y. Wu et C. Chen | Guiyang City, Guizhou Province, China |
| 25 | HS01 | Hushehong | *Ardisia mamillata* Hance | Ceheng County, Qiannan Prefecture, Guizhou Province, China |
| 26 | HS02 | Hushehong | *Ardisia mamillata* Hance | Congjiang County, Qiandongnan Prefecture, Guizhou Province, China |
| 27 | LZ01 | Lianzuozijinniu | *Ardisia primulifolia* Gardner & Champ. | Ceheng County, Qiannan Prefecture, Guizhou Province, China |
| 28 | XB01 | Xibingbailingjin | *Ardisia crispa* (Thunb.) A. DC. var. dielsii (Levl.) Walker | Changshun County, Guiyang City, Guizhou Province, China |
| 29 | XB02 | Xibingbailingjin | *Ardisia crispa* (Thunb.) A. DC. var. dielsii (Levl.) Walker | Qixingguan District, Bijie City, Guizhou Province, China |
| 30 | ZJ01 | Zijinniu | *Ardisia japonica* (Thunb.) Blume | Dushan County, Qiannan Prefecture, Guizhou Province, China |
| 31 | ZJ02 | Zijinniu | *Ardisia japonica* (Thunb.) Blume | Wulong District, Chongqing City, China |
| 32 | ZS01 | Zhushagen | *Ardisia crenata* Sims | Xixiu District, Anshun City, Guizhou Province, China |
| 33 | ZS02 | Zhushagen | *Ardisia crenata* Sims | Changshun County, Guiyang City, Guizhou Province, China |
| 34 | ZS03 | Zhushagen | *Ardisia crenata* Sims | Ceheng County, Qiannan Prefecture, Guizhou Province, China |
| 35 | ZS04 | Zhushagen | *Ardisia crenata* Sims | Yingjiang County, Yunnan Province, China |
| 36 | ZS05 | Zhushagen | *Ardisia crenata* Sims | Yingjiang County, Yunnan Province, China |
| 37 | ZS06 | Zhushagen | *Ardisia crenata* Sims | Guiyang City, Guizhou Province, China |
| 38 | ZS07 | Zhushagen | *Ardisia crenata* Sims | Ceheng County, Qiannan Prefecture, Guizhou Province, China |
| 39 | ZS08 | Zhushagen | *Ardisia crenata* Sims | Dushan County, Qiannan Prefecture, Guizhou Province, China |
| 40 | ZS09 | Zhushagen | *Ardisia crenata* Sims | Dushan County, Qiannan Prefecture, Guizhou Province, China |
| 41 | ZS10 | Zhushagen | *Ardisia crenata* Sims | Qianxi County, Bijie City, Guizhou Province, China |
| 42 | ZS11 | Zhushagen | *Ardisia crenata* Sims | Huishui County, Qiannan Prefecture, Guizhou Province, China |
| 43 | ZS12 | Zhushagen | *Ardisia crenata* Sims | Guiyang City, Guizhou Province, China |
| 44 | ZS13 | Zhushagen | *Ardisia crenata* Sims | Sandu County, Qiannan Prefecture, Guizhou Province, China |
| 45 | ZS14 | Zhushagen | *Ardisia crenata* Sims | Jichang Township, Xixiu District, Anshun City, Guizhou Province, China |
| 46 | ZS15 | Zhushagen | *Ardisia crenata* Sims | Ceheng County, Qiannan Prefecture, Guizhou Province, China |

Monte Carlo (MCMC) iterations after burn-in to 100,000 with a 100,000-run length, and the number of genetically homogeneous clusters (K value) ranged from 1 to 20 with 10 replicate runs for each analysis. The optimum K-value was determined by the highest K method in Structure Harvester3. The structure plot was constructed in R 4.1.0. The optimal set of core germplasm was extracted by the Core Hunter 3 (*Wang et al., 2023*), which maximized the genetic variation and allelic richness using local search algorithms. Based on the previously reported distribution of core germplasm fractions in woody plants ranging from 10 to 45%, we decided to test 10 sampling fractions (10, 15, 20, 25, 30, 35, 40, 45, and 50% and initial group) respectively by Core Hunter 3. As described above, *Na*, *Ne*, *Ho*, *He*, *I*, and *uHe* were calculated separately for each fraction using GenAlEx software. Using Microsoft Excel, these indicators were t-tested between the core subset and the initial group. The smallest core subset that did not differ significantly from the 100% population group ($P \leq 0.05$) was then selected as the optimal core germplasm collection.

# RESULTS

## Identification of frequency and distribution of EST-SSRs
### RNA sequencing, de novo assembly, and functional annotation

The same *Radix Ardisia* seedlings were selected for different light-intensity treatments, and the transcriptome was sequenced after sampling. Sequencing generated 141.85 G of data in total, and 71.62 GB and 70.23 GB of sequencing data (raw data) were obtained in the treatment group (intense light) and the control group (shading treatment), respectively. After removing the connector, the reads with NS (unable to determine the base information) ratio greater than 10% and low-quality reads (the base number with mass value *Qphred* $\leq 20$ accounts for more than 50% of the whole reads) are filtered. Finally, 66.67 GB and 66.65 GB of clean data are obtained for the treatment group (low temperature, 4 °C) and the control group (average temperature, 25 °C).

## Development and validation of the EST-SSR markers

In this study, 51,237 sequences were retrieved (Table S1), with a total length of 71.4 MB. A total of 32,827 SSR loci were detected (Table S2), averaging one SSR locus per 2.1 KB. The distribution of primers was detected. 28,322 SSR loci were unigene SSR, of which 9,455 were located in the 3′-UTR (Untranslated regions) region, accounting for 28.77%; 9,558 SSR loci were located in the CDS (Coding DNA sequence) region, accounting for 29.12%; 9,309 SSR loci were located in the 5′-UTR region, accounting for 28.36% (Table S3). 32,827 pairs of genome-wide EST-SSR markers were developed (Table S4). On average, there were 0.64 pairs of EST-SSR primers per unigene, and the sequence coverage was high. The statistical analysis results show that (3.1), six different types of nucleotide SSRs can be detected, but the number and frequency of SSRs in different primitive types vary greatly (Table 2). Mononucleotide and dinucleotide were the main repeat types, accounting for 90.51% of the total SSR. Single nucleotide SSRs were 14,692 pairs, accounting for 44.76% of the total SSRs, including 7,465 for (A), 7,043 for (T), 106 for (G), and 78 for (C), mainly of type A and T (Fig. 1); Class A accounts for 50.81% of the total number of this type, and class T accounts for 47.93%. Secondly, the frequency of dinucleotide repetition

**Table 2  Distribution characteristics of repeat motif types in the whole genome.**

| Type | Annotation | No. | Proportion |
|------|-----------|-----|-----------|
| P1 | Single nucleotide repeats | 14,692 | 44.76% |
| P2 | Dinucleotide repeats | 15,019 | 45.75% |
| P3 | Trinucleotide repeats | 2,337 | 7.12% |
| P4 | Tetranucleotide repeats | 544 | 1.66% |
| p5 | Pentanucleotide repeats | 109 | 0.33% |
| p6 | Hexanucleotide repeat | 126 | 0.38% |
| Total | | 32,827 | 100% |

also accounts for a large proportion. The number of dinucleotide SSRs is 15,019 pairs, accounting for 45.75%. The main types of dinucleotides are AG, AT, CT, GA, TA and TC, of which (AG) has 2,628, accounting for 17.50% of the total type, (AT) has 2,282, accounting for 15.19%, (CT) has 1707, accounting for 11.37%, (GA) has 2,380, accounting for 15.85%, (TA) has 2,128, accounting for 14.17%, (TC) has 2,348, accounting for 15.63%. There are 2,337 pairs of trinucleotide SSRs, accounting for 7.12%, including 204 (AAT), 100 (GAT), 138 (TTA) and 101 (ATT). The number of tetranucleotide, pentanucleotide and hexanucleotide repeat types is relatively small, which are 544 pairs, 109 pairs and 126 pairs, respectively, accounting for 1.66%, 0.33% and 0.38% of the total SSR (Fig. 2). There are 55 main types of tetranucleotides (ATAC) and 53 tetranucleotides (AAAT). The main types of pentanucleotide are (AAAAT) and (CCAAA), with six, respectively, and the main types of hexanucleotide are (GAGAGG) with three.

## Genetic diversity analysis

Sixty pairs of primers (Table S5), such as SSR1, SSR4, and SSR5, were randomly selected from the whole genome of *Radix Ardisia*, and PCR amplification was carried out for it. A total of 200 bands (Table S6) were amplified by 51 primers, with an average of 3.92 bands amplified by each primer, and the polymorphism rate (PPL) was 100%. The results of genetic diversity analysis showed that the genetic similarity coefficient between samples ranged from 0.6200 to 0.9350. At the species level, the number of polymorphic loci in the genus *Ardisia* was 200, the percentage of polymorphic loci (PPL) was 100.00%, and the total gene diversity ($Ht$) was 0.2181. The average number of observed alleles ($Na$) was 2.0000; the average number of effective alleles ($Ne$) was 1.3359; the average Nei's gene diversity index ($H$) was 0.2181; the average Shannon's polymorphism information index ($I$) was 0.3550 (Table 3). The results showed that 46 species of *Ardisia* had abundant genetic diversity and genetic variation among species and within species.

The principal component cluster analysis results of 46 samples of *Radix Ardisia* were obtained. The individuals were clustered into three dimensions by principal coordinates analysis (PCoA) (Fig. 2). PC1 represents the first principal component, PC2 represents the second principal component, and PC3 represents the third principal component. The variance contribution rates of PC1, PC2, and PC3 were 10.3%, 8.9%, and 7.07%, respectively. The population of Sapindus was split into four clusters, which corresponded approximately to the ZS group (Zhushagen, *Ardisia crenata* Sims), the HL group

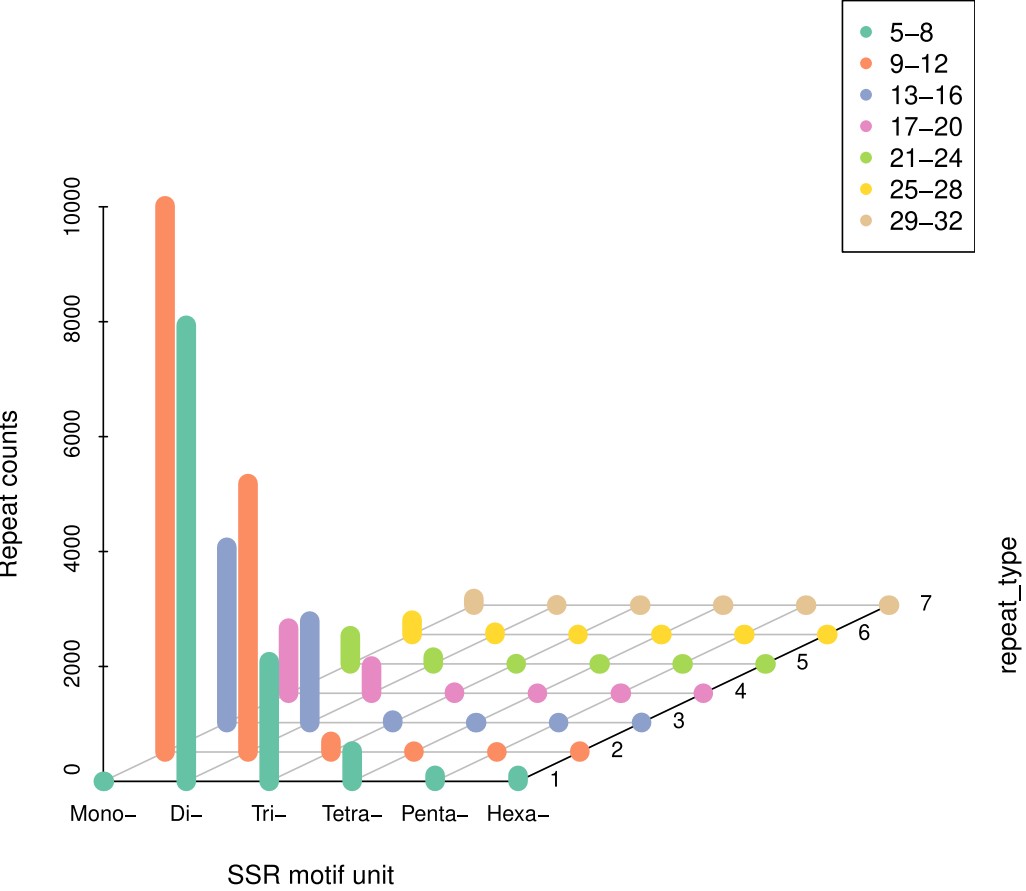

**Distribution of SSR Motifs**

**Figure 1** **Distribution characteristics of repeat motif types in the whole genome.**

(Hongliangsan, *Ardisia crenata* Sims var. bicolor (Walker) C. Y. Wu et C. Chen), the BH group (Baihuazijinniu, *Ardisia merrillii* E. Walker), and the BL (Bailiangjin, *Ardisia crispa* (Thunb.) A. DC.), DF (Dongfangzijinniu, *Ardisia elliptica* Thunb.), GG (Jiuguanxue, *Ardisia brevicaulis* Diels), HS (Hushehong, *Ardisia mamillata* Hance), LZ (Lianzuozijinniu, *Ardisia primulifolia* Gardner & Champ.), XB (Xibingbailingjin, *Ardisia crispa* (Thunb.) A. DC. var. dielsii (Levl.) Walker), ZJ (Zijinniu, *Ardisia japonica* (Thunb.) Blume) group.

Based on the EST-SSR primers developed, the population structure was analysed through the Structure software. In contrast to the PCoA results, when $K = 2$, the error rate of cross-validation is the minimum, so the optimal number of clusters is 2. 46 germplasm resources are divided into two subgroups, which shows a large genetic difference between the species, forming a significant genetic differentiation of the population (Fig. 3). When

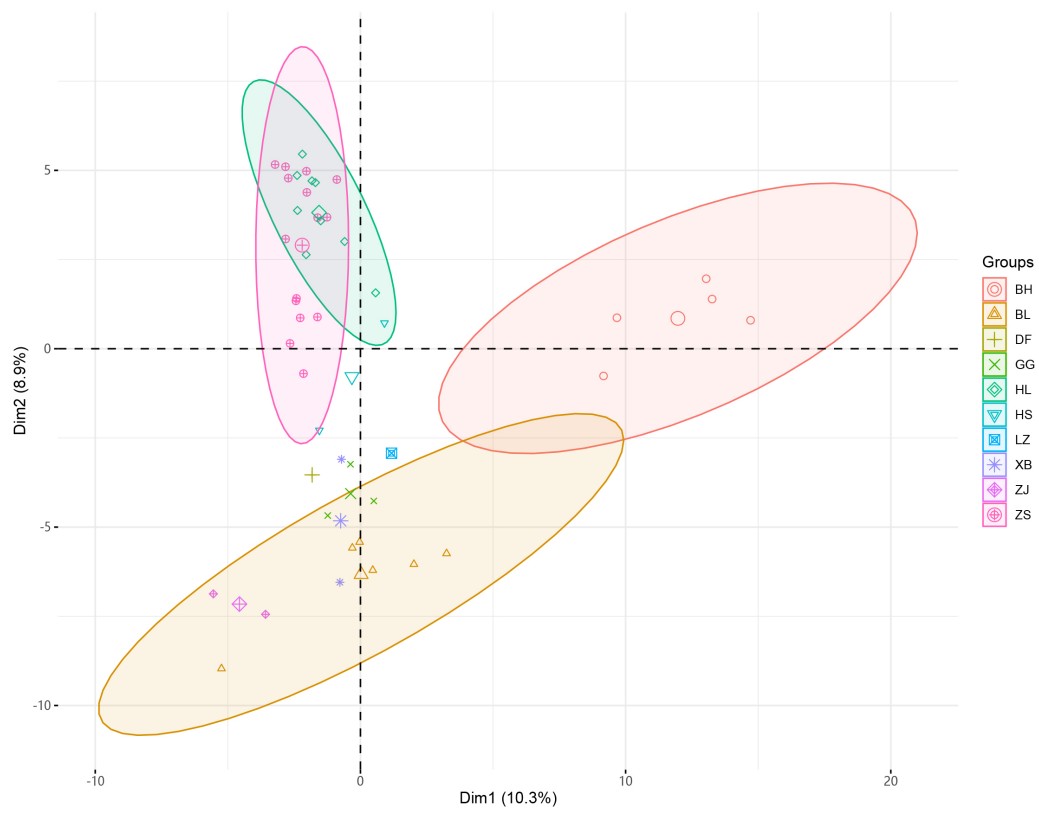

**Figure 2** PCA analysis of 46 samples of *Radix Ardisia*.

**Table 3** Genetic diversity analysis of *Radix Ardisia*.

| Locus | Sample size | *na* | *ne* | *h* | *I* | *Ht* |
|---|---|---|---|---|---|---|
| Mean | 46 | 2 | 1.3359 | 0.2181 | 0.355 | 0.2181 |
| St. Dev | | 0 | 0.2995 | 0.1516 | 0.1968 | 0.023 |

K was equal to 3–6, individuals of subgroup 2 were consistently divided into several subgroups.

The cluster analysis (Table S7) showed that *Ardisia*'s genetic distance ranged from 0.106 to 1.242. The results showed that EST-SSR could accurately distinguish the species of *Ardisia* and its relatives (Fig. 4); Zhushagen (*Ardisia crenata* Sims) and Hongliangsan (*Ardisia crenata* Sims var. bicolor (Walker) C. Y. Wu et C. Chen) are gathered into one branch, and Bailiangjin, Xibingbailiangjin, Dayebailiangjin, Zijinniu (*Ardisia japonica* (Thunb.) Blume), Baihua Zijinniu (*Ardisia merrillii* E. Walker), Dongfang Zijinniu (*Ardisia elliptica* Thunb.), Jiuguanxue (*Ardisia brevicaulis* Diels), and Hushehong (*Ardisia mamillata* Hance) are gathered into one branch respectively. Bailiangjin, Zijinniu, Baihua Zijinniu and Jiuguanxue can be clearly distinguished, and they are far away from Zhushagen and Hongliangsan, indicating that they are closely related, and their differentiation has prominent regional characteristics. At the same time, the genetic distance between Zhushagen, Hongliangsan

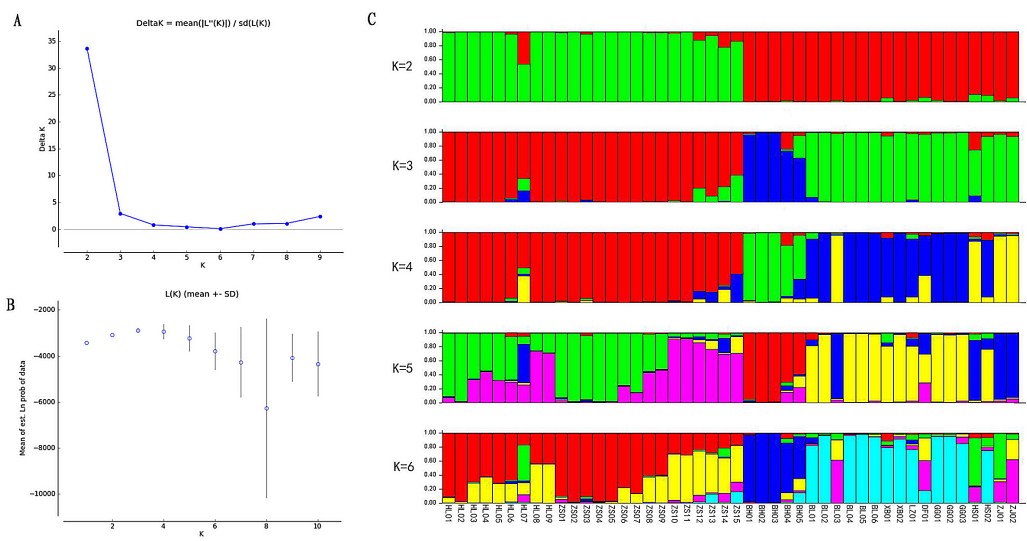

**Figure 3** Population structure analysis of *Radix Ardisia*.

and other species is far, and the species can be clearly distinguished. This shows that the EST-SSR used in the research can distinguish the *Radix Ardisia* and its easily mixed species of *Ardisia*, which can be applied to the identification. However, other data must be explored to distinguish Zhushagen and Hongliangsan.

## DISCUSSION

As a natural attribute formed in organisms' long-term evolution and development, genetic diversity is reflected not only among populations and within populations but also among individuals, including population and individual genetic variation (*Wang et al., 2023*). The genetic diversity of a species is not only the result of its long-term survival but also the result of evolution and adaptation (*Parveen et al., 2022*). Generally, the average value of the nucleotide diversity index of inbred species is 0.51, and that of outcrossing species is less than 0.1 (*Wen, Han & Wu, 2010*). In this study, the nucleotide diversity index of Ardisia is less than 0.1, which indicates that the cinnabar root is mainly cross-pollinated, and most of the genetic variation exists between populations.

The *Radix Ardisia* is the dried root and rhizome of Zhushagen, Hongliangsan and Bailiangjin. In the new edition of Flora of China, Zhushagen and Hongliangsan are combined into one species. In terms of appearance, the two are different in red or purple red. However, in the process of market use, the root is usually used, but the root cannot be identified through its appearance. Therefore, it needs to be accurately identified by microscopy (*Chen, Tong & Zhong, 2020*), molecular identification (*Zhang, Xing & Xue, 2021*; *Wei et al., 2020*) and other technologies; in this experiment, the genetic diversity of the three original plants of the plant were studied by EST-SSR markers. The intraspecific genetic distance of the three original plants was less than the interspecific genetic distance of the plant of the same genus, which showed significant differences between the medicinal

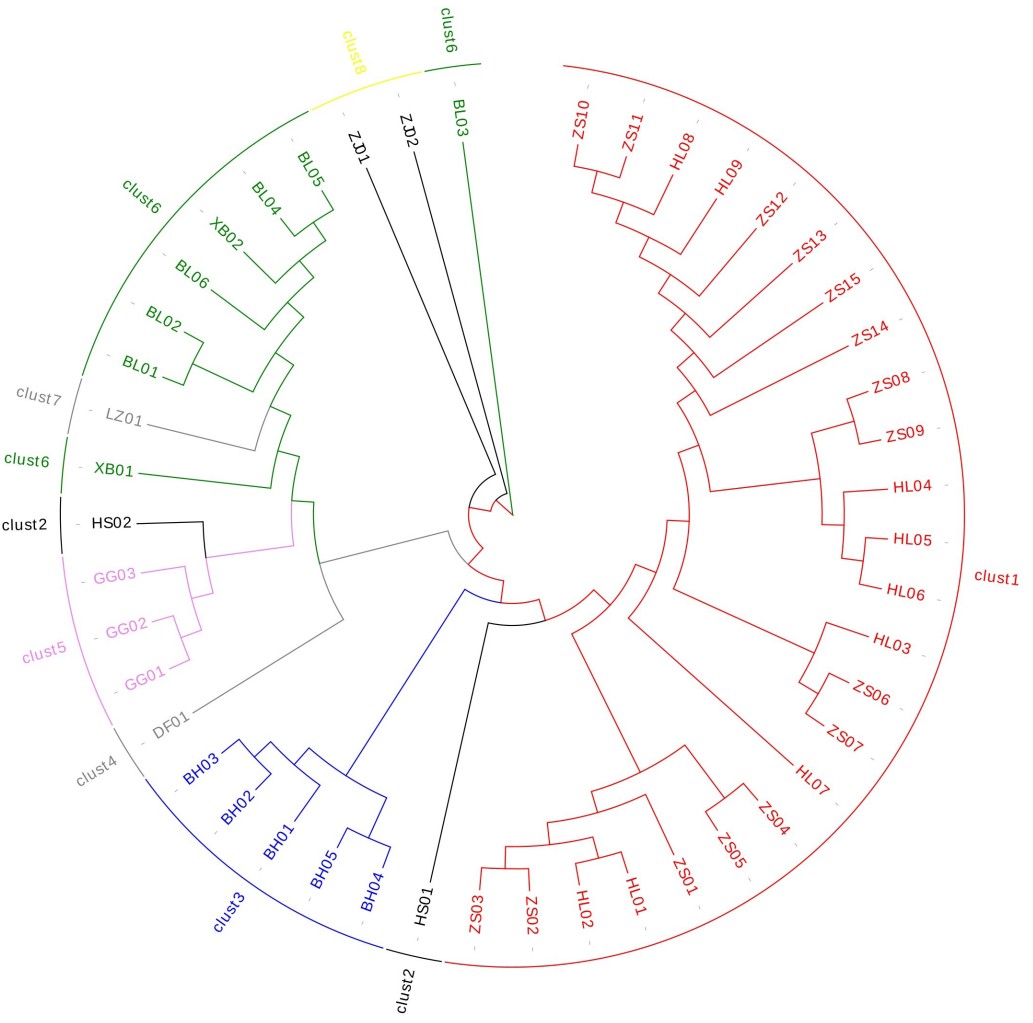

**Figure 4** **The cluster analysis of *Radix Ardisia*.**

material of the plant of the same genus at the molecular level. It could be well separated by K2P genetic distance. At the same time, the evolution tree constructed by the neighbour-joining method showed that the medicinal materials of *Radix Ardisia* could be effectively identified from other species of the same genus. However, Zhushagen and Hongliangsan can not be well distinguished, possibly because Hongliangsan is a variety of Zhushagen. There is a crossover phenomenon in genetic evolution.

In this study, transcriptome data used to develop EST-SSR molecular markers for genetic research can be better used in population structure analysis and identification research. EST-SSR is a powerful tool for genetic analysis. In addition to the advantages of high polymorphism, codominant inheritance, simple technology and good repeatability in gSSR markers, EST-SSR markers also have many other advantages. For example, EST-SSR markers can reduce the cost of primer development and significantly improve the utilization efficiency of existing sequencing data, making marker development more economical and

practical. Expressed sequence tag (EST) is a partial sequence of cDNA, representing a partial sequence of a complete gene. Therefore, SSR markers contained in EST can be used as direct markers of specific traits or genes. At the same time, EST-SSR flanking sequences are highly conservative and have good universality among different species of the same genus. Therefore, EST-SSR markers can be developed for those species with less research, less EST sequence information, or no EST data from their relatives. Developing molecular markers using EST sequences is a better way to develop and utilize EST databases. With the continuous enrichment of EST databases, developing SSR markers using EST sequences has become a simple and effective method. The development and research technology of EST-SSR marker is relatively mature and widely used in plants.

The EST-SSR sequence is a part of the coding gene, and its flanking sequence is highly conservative, so the primers designed can often be used for other species of the same genus or even family, which has been reported in many Rosaceae plants (*Preethi et al., 2024*; *Lu et al., 2021*; *Liu et al., 2022*). With 60 pairs of primers, 51 pairs of primers were found to have a good polymorphism in the amplification of 46 germplasm materials of *Ardisia*, indicating that the SSR flanking sequence in the target gene (target band) amplified by the 51 pairs of primers was highly conservative, which could not only be used for the study of the related molecular of *Radix Ardisia* but also be used for the further study of germplasm identification and genetic diversity analysis of other Ardisia plants. In addition, although the target fragment (main band) amplified by 51 pairs of primers is consistent with the expected product size, it may also contain false positive or amplification fragments without SSR repeat sequence, which must be sequenced.

Based on the extensive capacity transcriptome library of Zhuashagen that has been constructed, 32,827 SSR loci were developed for the first time through bioinformatics methods, with an average of one SSR locus every 2.1 Kb. This indicates that the EST sequence in *Radix Ardisia* contains rich repeats like other conventional crops, which provides an essential basis for identifying SSR in traditional Chinese medicine crops; it provides a basis for enriching the application of SSR in the role of traditional Chinese medicine. The dinucleotide content is the highest among the SSR sites developed in the selected dinucleotide repeats. The number of AG/GA/TC is the largest in different repeat units, consistent with the results of most species studies. This study also analysed the distribution characteristics of EST-SSR, which showed that 28.77% of EST-SSR was located in 3′-UTR, 28.36% of SSR was located in 5′-UTR and 29.12% of SSR was located in the CDS region, which was consistent with the distribution characteristics of EST-SSR in gene positions found in the study of dicotyledons by Kumpatla and Mukhopadhyay. It can be seen that EST-SSR has a strong conservatism in biological evolution.

Due to the lack of effective protection measures for the germplasm resources of *Ardisia* in China for a long time, there is still no unified standard for the effective division of varieties and germplasm resources in China. This is precisely why many cultivated varieties or wild germplasm resources containing excellent genetic information in the production of *Ardisia* are gradually lost and not used. It is difficult to distinguish or identify many varieties using traditional morphological identification methods. The existing markers, such as RAPD and *ITS2*, have been identified as effective in *Rehmannia glutinosa*. However, because it

is difficult to distinguish the effective heterozygotes of germplasm, and these markers have complex steps and relatively high requirements, their promotion and utilisation in growth practice are affected. Therefore, there is an urgent need for a modern molecular marker method to carry out the most fundamental differentiation and identification from its "internal" way. Among the molecular markers used, only SSR is the most simple and economical way. In Zijinniu, due to the fuzzy genomic information and the few available genetic resources, the application of SSR in variety breeding is minimal; this research is the first large-scale development of practical EST-SSR markers with polymorphism of *Ardisia* based on the transcriptome. At the same time, this research is also the first to carry out preliminary systematic genetic clustering of existing germplasm resources, which will play an important role in the subsequent cultivar cultivation and identification of germplasm resources.

## CONCLUSIONS

The results of systematic clustering showed that the EST-SSR used could well distinguish *Radix Ardisia* and its easily mixed species; it can be applied to the identification of *Radix Ardisia*, as well as the molecular identification between the original Bailiangjin, Zhushagen and Hongliangsan. This study can provide a reference for the genetic analysis of the *Radix Ardisia*.

### Funding

This research was funded by the Post subsidy Special Project of National Natural Science Foundation of China "Study on the Mechanism of the Effect of Light on the Growth and Development and the Quality Formation of Miao medicine Radix Ardisia" (Grant No. 2018YFC170810104). The funders had no role in study design, data collection and analysis, decision to publish, or preparation of the manuscript.

### Grant Disclosures

The following grant information was disclosed by the authors:
Post subsidy Special Project of National Natural Science Foundation of China: 2018YFC170810104.

### Competing Interests

The authors declare there are no competing interests.

### Author Contributions

- Deqiang Ren conceived and designed the experiments, analyzed the data, prepared figures and/or tables, authored or reviewed drafts of the article, and approved the final draft.
- Wenwen Wu performed the experiments, analyzed the data, authored or reviewed drafts of the article, and approved the final draft.

- Qinqin Wen performed the experiments, prepared figures and/or tables, authored or reviewed drafts of the article, and approved the final draft.
- Yujiao Li performed the experiments, prepared figures and/or tables, and approved the final draft.

## Data Availability

The data is available at NCBI: GLBD00000000.1, PRJNA1199703, SAMN45885934.

## Supplemental Information

Supplemental information for this article can be found online at http://dx.doi.org/10.7717/peerj.19560#supplemental-information.

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
