# Peer review of "Development of EST-SSR markers based on transcriptome for genetic analysis in Radix Ardisia"

_PeerJ, doi:10.7717/peerj.19560_

## Round 0.1 · original submission · Major Revisions

The authors are requested to carefully revise the manuscript and answer the questions raised by the reviewers.

**Language Note:** The review process has identified that the English language must be improved. PeerJ can provide language editing services - please contact us at [email protected] for pricing (be sure to provide your manuscript number and title). Alternatively, you should make your own arrangements to improve the language quality and provide details in your response letter. – PeerJ Staff

Reviewer 1 ·

Basic reporting

In this research paper, it has been aimed to develop genome-wide EST-SSR primers using transcriptome data of Ardisia crenata Sims species. It has been emphasized that the developed markers would be useful in research topics such as genetic diversity of Ardisia species, population genetic structure, and identification of germplasm resources.
English language usage is sufficient.

Literature references are sufficient.

Figures and tables are sufficient. However, references to "supplementary materials" are not specified in the relevant sections of the text.

Figures are sufficient. However, "Among the clusters highlighted with various colors in Figure 4, "cluster8" is shown in yellow on the figure. However, it can be read when the figure is enlarged".

Results compatible with the purpose of the study were determined.

Experimental design

It has been determined that the experimental design were well prepared.
However, some of the identified deficiencies have been listed below.
1. Under the Introduction heading (Page 7 line 53), Radix Ardisia genetic diversity is emphasized by giving reference sources numbered 4, 5 and 6. However, according to the sources given in this sentence, it would have been more accurate to have the species name information “Ardisia crenata”.
2. Under the Introduction heading (on page 7, line 71), in the sentence explaining the purpose of the research, it is shared that the species name “Ardisia crenata Sims” is the basic plant material, but this species information is not given under the material method heading. Additionally, no reference is made in the text for supplementary materials.
3. The number of plant material samples used in the study is not given under the heading “Experimental materials”. The number of samples given on page 9, line 158 (46 samples) would have been appropriate to be given under the subheading “Experimental Materials” under the heading “Materials & Methods”.
4. The word “s equencing” in the heading on line 82 of page 7 should be corrected.
5. The number of primers and samples in the sentence “With 60 pairs of primers, 51 pairs of primers were found to have a good polymorphism in the amplification of 46 germplasm materials of Ardisia, …” on page 11, lines 217-218 should have been given under the subheading “Experimental materials”.
6. Similar determination is available. The information about 51 pairs of primers in the sentences “Sixty pairs of primers were randomly selected and applied to genetic research. Among them, 51 pairs could amplify 200 polymorphic bands” given in the abstract (page 6 line 32) and under the results heading (page 9 line 149) should have been shared.
7. Citation information for reference 23 could not be found in the text.
8. The ITS2 marker name on page 11, line 242 is written in italics and should be corrected.
9. Abbreviations could be explained in parentheses at their first mention, for example, EST-SSR, CBS, UTR, etc.

Validity of the findings

There is an unclear finding regarding the validity of the research findings.

Additional comments

The information given in the sentence on line 91 of Page 8, “accession number SUB19947097” was searched on the NCBI web page. However, no information was found regarding the genus Ardisia (02/09/2025-02/13/2025) (https://www.ncbi.nlm.nih.gov/search/all/?term=14947097). This is not understood.

Reviewer 2 ·

Basic reporting

In this manuscript, the authors developed EST-SSR markers from the Radix Ardisia transcriptome and analyzed the genetic diversity of different Radix Ardisia materials. Although some genetic analysis was provided, the logic of this manuscript is confusing, and I found many errors, including the results, expression and grammar in the manuscript.

Experimental design

1. Please provide detailed information on the collected samples.
2. Lines 147-157, please provide the electrophoresis results to support the polymorphism of the 60 pairs of primers selected.

Validity of the findings

1. Line 123-127, "In this study, 51237 sequences were retrieved, with a total length of 71.4MB. A total of 32827 SSR loci 124 were detected, averaging 1 SSR locus per 2.1 KB. The distribution of primers was detected. 28322 SSR 125 loci were unigene SSR, of which 9455 were located in the 3'-UTR region, accounting for 28.77%; 9558 126 SSR loci were located in the CDS region, accounting for 29.12%; 9309 SSR loci were located in the 5 '- 127 UTR region, accounting for 28.36%." Please provide the source of the data.
2. It is suggested to replace lines 158-162 with a description of the PCA results, which should include the information within each cluster.
3. In Figure 3, based on the detal K value, the filtered K value should be 2/3/9 instead of 6. Correspondingly, the conclusion in lines 163-166 is wrong.
4. The genetic similarity coefficient can not be concluded from Figure 4.

Additional comments

I found too many formatting and expression errors in this manuscript.

---

## Round 0.2 · accepted · Accept

After revisions, one reviewer agreed to publish the manuscript. I also reviewed the manuscript and found no obvious risks to publication. Therefore, I also approved the publication of this manuscript.

Reviewer 1 ·

Basic reporting

The corrected article text has been reconsidered. It has been determined that my suggestions have been corrected by the authors. The results of this study may be helpful for the identification of Ardisia germplasm resources and early selection studies.

Experimental design

This research article has been determined as an original article. The research purpose, method and results have been found to be compatible.

Validity of the findings

The results of this study may be helpful for the identification of Ardisia germplasm resources and early selection studies.